A predictive model for canine dilated cardiomyopathy—a meta-analysis of Doberman Pinscher data

Simpson Siobhan 1
Edwards Jennifer 1
Emes Richard D. 1 2
Cobb Malcolm A. 1
Mongan Nigel P. 1 3 Nigel.mongan@nottingham.ac.uk npm2001@med.cornell.edu
Rutland Catrin S. 1
1 School of Veterinary Medicine and Science, Sutton Bonington Campus, University of Nottingham , Leicestershire , UK
2 Advanced Data Analysis Centre, Sutton Bonington Campus, University of Nottingham , Leicestershire , UK
3 Department of Pharmacology, Weill Cornell Medical College , New York, NY , USA
Esteban María Ángeles
Electronic publication date: 2015 Mar 26
Publication date: 2015
Volume: 3
Electronic Location ID: e842
Received 2014 Nov 14; Accepted 2015 Feb 27
Copyright: © 2015 Simpson et al.
Copyright year: 2015
Copyright holder: Simpson et al.
License: This is an open access article distributed under the terms of the Creative Commons Attribution License, which permits unrestricted use, distribution, reproduction and adaptation in any medium and for any purpose provided that it is properly attributed. For attribution, the original author(s), title, publication source (PeerJ) and either DOI or URL of the article must be cited.
License URL: https://creativecommons.org/licenses/by/4.0/

Keywords: Dilated cardiomyopathy, PDK4, Canine, Multigenic, Human

Funding: Biotechnology and Biological Sciences Research Council (BBSRC) BB/J014508/1 School of Veterinary Medicine and Science University of Nottingham Doctoral Training Program Boehringer Ingelheim Vetmedica This work was funded by the Biotechnology and Biological Sciences Research Council (BBSRC) BB/J014508/1 and the School of Veterinary Medicine and Science, University of Nottingham Doctoral Training Program, and in part by an unrestricted educational research grant provided by Boehringer Ingelheim Vetmedica to Catrin S. Rutland, Nigel P. Mongan and Malcolm Cobb. The funders had no role in study design, data collection and analysis, decision to publish, or preparation of the manuscript.

==============================
Dilated cardiomyopathy is a prevalent and often fatal disease in humans and dogs. Indeed dilated cardiomyopathy is the third most common form of cardiac disease in humans, reported to affect approximately 36 individuals per 100,000 individuals. In dogs, dilated cardiomyopathy is the second most common cardiac disease and is most prevalent in the Irish Wolfhound, Doberman Pinscher and Newfoundland breeds. Dilated cardiomyopathy is characterised by ventricular chamber enlargement and systolic dysfunction which often leads to congestive heart failure. Although multiple human loci have been implicated in the pathogenesis of dilated cardiomyopathy, the identified variants are typically associated with rare monogenic forms of dilated cardiomyopathy. The potential for multigenic interactions contributing to human dilated cardiomyopathy remains poorly understood. Consistent with this, several known human dilated cardiomyopathy loci have been excluded as common causes of canine dilated cardiomyopathy, although canine dilated cardiomyopathy resembles the human disease functionally. This suggests additional genetic factors contribute to the dilated cardiomyopathy phenotype.This study represents a meta-analysis of available canine dilated cardiomyopathy genetic datasets with the goal of determining potential multigenic interactions relating the sex chromosome genotype (XX vs. XY) with known dilated cardiomyopathy associated loci on chromosome 5 and the PDK4 gene in the incidence and progression of dilated cardiomyopathy. The results show an interaction between known canine dilated cardiomyopathy loci and an unknown X-linked locus. Our study is the first to test a multigenic contribution to dilated cardiomyopathy and suggest a genetic basis for the known sex-disparity in dilated cardiomyopathy outcomes.

Introduction

Dilated cardiomyopathy (DCM) is a prevalent and often fatal disease requiring clinical management in humans and dogs (Egenvall, Bonnett & Häggström, 2006; Hershberger, Morales & Siegfried, 2010). DCM is the second most common cardiac disease in dogs and is characterised by ventricular chamber enlargement and systolic dysfunction which often leads to congestive heart failure (Egenvall, Bonnett & Häggström, 2006). The aetiology of DCM is complex. Genetic factors, myocardial ischemia, hypertension, toxins, infections and metabolic defects have been implicated (McNally, Golbus & Puckelwartz, 2013). To date, mutations in over 50 genes have been associated with DCM in humans; however, mutations in the most prevalent DCM related genes only account for approximately 50% of patients with DCM (Posafalvi et al., 2013). In human DCM genetic testing where a panel of approximately 50 loci are tested concurrently, often more than one locus can be implicated in the disease (McNally, Golbus & Puckelwartz, 2013), suggesting multiple genetic factors cooperate in DCM aetiology.

Canine DCM is phenotypically similar to human DCM (Shinbane et al., 1997). As outlined below, to date mutations in only two genes (PDK4 and STRN) and a single nucleotide polymorphism (SNP) on chromosome 5 have been associated with canine DCM (Mausberg et al., 2011; Meurs et al., 2012; Meurs et al., 2013), suggesting additional genetic causes remain unknown. While canine studies have sometimes been limited by small sample size (typically less than 10 individuals), those studies with larger sample numbers (greater than 50 individuals) have also frequently failed to find significant associations with DCM (e.g., Philipp et al., 2007; Philipp, Vollmar & Distl, 2008; Wiersma et al., 2008). One possible explanation for the challenges in identifying DCM associated loci in humans and dogs is that even within an extended family or breed, genetic variation at a single locus cannot explain the development of DCM. Indeed dog breeds can be considered as large families, with dogs within a breed more related to each other than dogs of other breeds (Parker et al., 2004). In the same way that some human families are affected by DCM, a subset of dog breeds are affected by DCM more frequently than others (Egenvall, Bonnett & Häggström, 2006). Dobermans Pinschers (hereafter Dobermans) are particularly affected by DCM, with both a high prevalence (58.2% in European Dobermans) and severity with DCM associated death often occurring within 8 weeks of diagnosis (Calvert et al., 1997; Wess et al., 2010). In dogs, diagnosis is usually at the onset of clinical symptoms of heart failure. But there is an extended pre-clinical phase, during which if treatment can be effective by prolonging the onset of heart failure (Summerfield et al., 2012). In this phase left ventricular dilation and dysfunction begins, and can be accompanied by ventricular premature complexes (Singletary et al., 2012), Median life expectancy of DCM affected European Dobermans is 7.8 years, compared with 11 years for unaffected European Dobermans (Proschowsky, Rugbjerg & Ersbøll, 2003; Egenvall, Bonnett & Häggström, 2006). A deletion in a splice site of the PDK4 gene (Meurs et al., 2012) and a SNP on chromosome 5 (Mausberg et al., 2011) in Dobermans are two of only three canine DCM mutations identified to date. While these two loci are associated with Doberman DCM, individually neither locus explains all cases of Doberman DCM (Mausberg et al., 2011; Meurs et al., 2012). Individuals heterozygous at the Chr5 SNP are more likely to develop DCM, but there are many DCM cases homozygous for the healthy allele (Mausberg et al., 2011). While PDK4 genotypes are less definite predictors of DCM, with both affected and unaffected individuals possessing the three possible genotypes, the 16bp PDK4 splice site deletion is found more frequently in North American Dobermans with DCM than those without DCM (Meurs et al., 2012). However an analysis of European Dobermans failed to identify an association between PDK4 and DCM (Owczarek-Lipska et al., 2013), suggesting additional unknown factors influence the effect of PDK4 in predisposing individuals to DCM. Thus novel genetic causes of canine DCM remain to be identified (Mausberg et al., 2011; Philipp et al., 2012).

In this study we developed genetic models to test the influence of unknown genetic factors to predict which DCM-associated genotype combinations are likely to develop DCM. Using this method, we provide evidence for a sex-linked genetic influence on known DCM loci in the pathogenesis of canine DCM. Our study is the first to propose a multigenic contribution to canine DCM.

Materials and Methods

Model development

A literature search of the Pubmed and Web of Science database using the following search terms: “Doberman DCM loci,” “Doberman Dilated Cardiomyopathy loci,” “Doberman DCM gene,” “Doberman Dilated Cardiomyopathy gene,” “Doberman DCM locus” and “Doberman Dilated Cardiomyopathy locus” (Fig. 1) identified thirty unique records. This search identified two loci associated with DCM in North American Dobermans (Meurs et al., 2012; Mausberg et al., 2011). By combining the genotypes from the identified Doberman DCM associated loci, and additional putative loci, predictive models were developed and tested against observed DCM incidence data. All genotype combinations for the DCM associated SNP identified on chromosome 5 (TIGRP2P73097:CFA5:g.53,941,386T>C, CanFam2.1) (Mausberg et al., 2011) and the PDK4 (GeneID:482310) splice site deletion (CFA14:g.20,829,667_20,829,682del, CanFam3.1) (Meurs et al., 2012) were determined. Further analysis determined which genotype combinations were likely to lead to DCM. Some genotypes are definitive; all individuals homozygous for the susceptibility allele at CFA5:g.53,941,386T>C develop DCM (Mausberg et al., 2011).

Figure 1 Prisma flow diagram.

Determining which genotypes develop DCM

Five genetic models incorporating genotypes at multiple observed and hypothetical loci were developed including: 1. two known DCM loci; 2. two known loci + 50% of the population more susceptible to developing DCM; 3. two known loci + a novel autosomal dominant DCM locus; 4. two known loci + a novel autosomal recessive DCM locus; 5. two known loci + a novel additive DCM locus and 6. two known loci + a novel X-linked DCM locus. For each model, different biologically feasible phenotype outcomes were tested for each genotype combination to establish the best fit of the model to the observed DCM incidence data. Each model was subject to the following constraints: individuals that are homozygous CC at the Chr5 SNP develop DCM, and individuals with no susceptibility alleles are healthy.

Model testing

For each model, the frequency of each genotype combination was calculated by multiplying the genotype frequencies using PDK4 and Chr5 frequencies (Table 1) obtained from Owczarek-Lipska et al. (2013) and Mausberg et al. (2011). A range of frequencies were tested for each hypothetical loci. For example, for the model incorporating only PDK4 and Chr5 variants, one genotype combination is WtWt-TT. The frequency of this genotype combination is the product of the frequency of WtWt and the frequency of TT in the population. From the combined genotype frequencies, the expected numbers of individuals with each genotype combination were calculated by multiplying the frequency by the number of individuals in the study to be compared with (182 when compared with Mausberg et al. (2011) and Owczarek-Lipska et al. (2013)). Thus, the numbers of individuals in the model that were, for example, WtWt healthy and WtWt DCM were obtained by summing the numbers in each category. Having obtained the numbers of affected and unaffected individuals that the model predicts for each genotype, these were tested against the observed data using a χ2 test. Where additional putative DCM loci were included in the model, several allele frequencies were tested. However, as GWAS studies have previously been carried out (Mausberg et al., 2011; Meurs et al., 2012), it is unlikely that additional DCM alleles are at higher frequencies than those already identified. For this reason, DCM allele frequencies over 0.5 were not tested. If the model is a good fit of the observed data, the χ2 test statistic will be non-significant.

Table 1 Genotype frequencies assuming Hardy Weinberg equilibrium.

Allele frequencies taken from Mausberg et al. (2011) and Owczarek-Lipska et al. (2013).

PDK4	Chr5 SNP	
Genotype	Freq	Genotype	Freq	
Wt Wt	0.72	TT	0.74	
Wt del	0.26	TC	0.24	
Del del	0.02	CC	0.02	

The proportion of the population that the model predicts to have DCM was determined by taking the sum of all the genotype combined frequencies that lead to DCM in the model. For example, for the model incorporating just the two known loci this is 0.0144 + 0.0624 + 0.0052 + 0.0048 + 0.0004 = 0.0872—(Table S1). This proportion was then compared to the observed DCM frequency of 0.582 (Wess et al., 2010).

For most models, it must be assumed that there is no difference in DCM incidence between the sexes, as an effect of sex has not been included. For the DCM model testing a 50% increased susceptibility, where it is biologically feasible that males are more susceptible and the models incorporate an additional X-linked locus, it is possible to calculate the proportion males and females that develop DCM. While males develop clinical symptoms earlier and appear to be more severely affected, there are indications that the sex of those affected by DCM is close to 50% male, 50% female (Wess et al., 2010), so we would expect our model to reflect this.

Odds ratios of each genotype and allele developing DCM for each model were obtained by testing each genotype against the other two combined and each allele against the other. Odds ratios are the odds/probability of an individual with a particular genotype or allele developing DCM compared, by dividing one by the other, to the odds of an individual with all other genotypes or alleles developing DCM, with and odds ratio greater than one associated with the trait of interest and an odds ratio of less than one not associated (Bland & Altman, 2000). For example the odds ratio for TT in the published data from Mausberg et al. (2011) is calculated in the following way. There are 45 individuals that are TT DCM and 85 TT healthy the odds of a TT individual developing DCM are 45/85 (0.53), there are 43 individuals which are TC or CC with DCM and 9 individuals that are TC or CC healthy so the odds of these individuals developing DCM are 43/9 (4.78) the odds ratio divides the genotype of interest odds by the ‘others’ odds to give the odds ratio or 0.11. To assess the significance of these ratios χ2 tests were performed on the 2 × 2 tables—in the above example the four groups are TT-DCM, TT-healthy, TC or CC-DCM, TC or CC-healthy. If the model is a good fit to the observed data it is expected that the odds ratios are of a similar pattern and significance, e.g., TT, small—significantly not associated with DCM; TC, large—significantly associated with DCM; CC, not possible to test—not testable, as for the Chr5 SNP in Table 2. Odds ratios of both genotypes and alleles were obtained from the original studies (Tables 2 and 3).

Table 2 Genotype odds ratios from the original studies reporting an association.

Ratios from the PDK4 locus (Meurs et al., 2012) and Chromosome 5 SNP (Mausberg et al., 2011). The PDK4 χ2 test results indicate that the WtWt genotype significantly associated with non-DCM and the WtDel genotype significantly associated with DCM at the 0.01 significance level, the DelDel genotype odds ratio whilst different from the null result of 1, is not significantly so. For the chromosome 5 SNP all individuals that are CC in the original study developed DCM, thus and odds ratio and confidence interval cannot be calculated, but χ2 tests can be performed on the data. TT is significantly associated with non-DCM and the TC and CC genotypes are significantly associated with DCM at the 0.01 significance level.

Genotype	Odds ratio	95% CI	
PDK4 WtWt	0.14	0.07, 0.32	
PDK4 WtDel	5.21	2.70, 12.09	
PDK4 DelDel	1.14	0.41, 3.18	
Chr5 TT	0.11	0.05, 0.24	
Chr5 TC	6.23	2.78, 14.00	
Chr5 CC	NA	NA	

Table 3 Allele odds ratios from the original studies reporting DCM associations.

Allele odds rations at the PDK4 locus (Meurs et al., 2012) and Chromosome 5 SNP (Mausberg et al., 2011). The χ2 test results indicate that each susceptibility (Del and C respectively) allele is significantly associated with DCM and the alternate allele significantly associated with non-DCM at the 0.01 significance level.

Allele	Odds ratio	95% CI	
PDK4 Wt	0.38	0.23, 0.64	
PDK4 Del	2.63	1.57, 4.42	
Chr5 T	0.12	0.06, 0.26	
Chr5 C	8.11	3.85, 17.09	

Results

Following the constraints stated in the methods and using biologically feasible reasoning each model was optimised to best fit the observed data. For each model the genotype-phenotype decision descriptions are shown in Table 4. Tables of each model are in Supplemental Information.

Table 4 Genotype-phenotype decision descriptions for each model.

Models represent: 1. the two known DCM loci; 2. two known loci + 50% of the population is more susceptible to developing DCM; 3. two known DCM loci combined with a novel autosomal dominant DCM locus; 4. two known DCM loci combined with an autosomal recessive locus; 5. two known DCM loci combined with a an additional DCM locus that is additive and 6. two known DCM loci combined with an X-linked DCM locus.

Model	Genotype—phenotype decision description, in addition to the rules:	
1.	DCM develops when both the PDK4 locus and Chr5 SNP have at least one DCM susceptibility allele.	
2.	50% more susceptible only need to have a single DCM susceptibility allele at either locus to develop DCM while the 50% less susceptible to DCM require at least one DCM susceptibility allele at both loci to develop DCM.	
3.	All individuals that have a susceptibility allele at the additional locus develop DCM. Those individuals with no susceptibility alleles at the additional locus need at least one DCM susceptibility allele at both of the other loci to develop DCM.	
4.	All homozygous susceptible individuals at the additional locus develop DCM. For individuals that are heterozygous at the additional locus, DCM occurs when combined with another DCM susceptibility allele, while homozygous unsusceptible individuals need at least one DCM susceptibility allele at both of the other loci to develop DCM.	
5.	All homozygous susceptible individuals at the additional locus develop DCM. Heterozygotes and homozygous unsusceptible individuals need at least one DCM susceptibility allele at both of the other loci to develop DCM.	
6.	X linked susceptible DCM locus males can either possess a single unsusceptible X (XY) or a single susceptible x (xY), while females can be unsusceptible X homozygotes (XX), heterozygotes (Xx) or susceptible x homozygotes (xx). Unsusceptible X males (XY) are phenotypically identical to unsusceptible X homozygotes (XX) with these individuals requiring at least one DCM susceptibility allele at both of the other loci to develop DCM. All individuals that possess a susceptible X (xY and xx individuals) develop DCM in this model while heterozygotes (Xx) only require a single DCM susceptibility allele at one of the other loci to develop DCM.	

Comparing model predictions with observed data

The χ2 test values comparing predicted numbers with observed numbers of DCM and healthy individuals at each genotype ranged from 4.35 to 7766.06. A χ2 value of less than 11.07 indicates there is no significant difference between predicted and observed genotype-phenotype data, (5% significance level, with 5 degrees of freedom). Values less than 15.09 represent predictions not significantly different to observed values at the 1% significance level. χ2 values less than these critical values are indicated in Table 5.

Table 5 χ2 test statistic results comparing predicted of DCM and healthy individuals at each genotype from each model with observed numbers of DCM and healthy individuals at each genotype.

Model data based on data from Mausberg et al. (2011)–Chr5 SNP and Owczarek-Lipska et al. (2013)–PDK4.

	χ2 test statistic for each model	
Model	PDK4				Chr5				
1.	1269.23				7766.06				
2.	110.45				596.68				
DCM allele freq	0.5	0.4	0.3	0.2	0.5	0.4	0.3	0.2	
3.	32.47	29.25	51.42	113.35	6.58**	7.69**	24.30	69.27	
4.	26.24	74.61	171.69	379.06	31.65	67.45	145.76	360.86	
5.	88.95	31.36	4.97**	4.36**	114.72	53.10	23.13	17.21	
DCM X allele (x) freq	0.5	0.4	0.3	0.2	0.5	0.4	0.3	0.2	
6.	10.57**	10.06**	25.38	71.30	11.32*	9.29**	19.55	52.86	
Notes.

* not significant at 1% significance level.

** not significant at 5% significance level.

Model predicted DCM population frequency and sex incidence

For each model, the predicted DCM frequency was calculated to provide an additional method to test the accuracy of the model. The DCM frequency in the European Doberman population is estimated to be 58.2% (Wess et al., 2010), therefore accurate models should predict a similar frequency. The frequencies predicted by each model are displayed in Table 6 (see also Table S2), with those within 10% of the reported frequency highlighted as accurate models. Further to this the proportion of males and females that each model predicts to develop DCM were calculated. Whilst most models do not account for sex and assume equal numbers of males and females affected, two models tested either a 50% increase in male susceptibility or an additional X-linked locus. Based on reported DCM incidence for a model to fit the observed data it is expected that similar proportions of males and females develop DCM. Table 7 shows that irrespective of the frequency of the novel susceptibility allele the model incorporating a novel X linked DCM locus gives similar proportions of affected males and females.

Table 6 DCM frequency predicted by each model.

Model	DCM freq for each model	
1.	0.0872				
2.	0.2772				
DCM allele freq	0.5	0.4	0.3	0.2	
3.	0.5054*	0.415648	0.328952	0.245321	
4.	0.3154	0.233248	0.169352	0.123712	
5.	0.7718	0.671392*	0.552728*	0.415808	
DCM X allele (x) freq	0.5	0.4	0.3	0.2	
6.	0.5245*	0.433984	0.350432	0.257536	
Notes.

* indicates frequencies within 0.1 of the reported frequency (0.582 (Wess et al., 2010)) in the European Doberman pincher population.

Table 7 Proportion of males and females predicted to be affected by DCM by models 2 and 6.

	Proportion DCM	
Model	Male	Female	
2.	0.4672	0.0872	
6.			
DCM X allele (x) freq 0.5	0.5436	0.5054	
DCM X allele (x) freq 0.4	0.45232	0.415648	
DCM X allele (x) freq 0.3	0.36104	0.339824	
DCM X allele (x) freq 0.2	0.26976	0.245312	

Odds ratios

For the Chr5 SNP there are no odds ratio for CC, as all individuals that are CC develop DCM in both the original study (Mausberg et al., 2011) and models so odds ratios cannot be calculated. Despite this a χ2 test can be performed on the counts of affected and unaffected individuals observed and predicted with the genotype so the significance of the results was obtained. For the Chr5 SNP, 12 of 18 models (Table 9), and 15 of the allele odds ratios are consistent with the original studies (Table 11). The PDK4 deletion association was identified in the North American Doberman population; in the European population, the odds ratios (Tables 8 and 10) are not significantly different from the null result of 1. Once combined with additional loci, similar significant likelihood ratios as the North American population are obtained for 13 of 18 models (Tables 8 and 10).

Table 8 Odds ratios of each PDK4 genotype with χ2 significance.

	PDK4 genotype odds ratio	
Model	wtwt	wtdel	deldel	wtwt	wtdel	deldel	wtwt	wtdel	deldel	wtwt	wtdel	deldel	
Individual loci	0.78	1.29	1.11										
1.	0.06**	12.91**	3.85										
2.	0.1**	9.41**	4.6										
DCM allele freq	0.5	0.4	0.3	0.2	
3.	0.14**	6.70**	4.42	0.15**	6.31**	3.98	0.15**	6.21**	3.69	0.14**	6.47**	3.53	
4.	0.45*	2.17*	1.76	0.35**	2.73**	2.03	0.25**	3.77**	2.43	0.15**	5.82**	2.98	
5.	0.7	1.42	1.31	0.67	1.49	1.36	0.62	1.6	1.43	0.53	1.84	1.58	
DCM X allele (x) freq	0.5	0.4	0.3	0.2	
6.	0.31**	3.12**	2.4	0.30**	3.23**	2.41	0.28**	3.41**	2.45	0.24**	3.89**	2.59	
Notes.

* significant at 5% level

** significant at 1% level

Table 9 Odds ratios of each Chr5 SNP genotype with χ2 significance.

Model	Chr5 genotype odds ratio	
	TT	TC	CC	TT	TC	CC	TT	TC	CC	TT	TC	CC	
Individual loci	0.11**	6.23**	–**										
1.	0.02**	11.37**	–**										
2.	0.09**	9.23**	–**										
DCM allele freq	0.5	0.4	0.3	0.2	
3.	0.14**	6.74**	–	0.14**	6.34**	–*	0.13**	6.25**	–**	0.12**	6.56**	–**	
4.	0.35**	2.33*	–**	0.25**	2.96**	–**	0.16**	4.13**	–**	0.08**	6.45**	–**	
5.	0.67	1.51	–	0.61	1.57	–	0.54	1.7	–	0.44*	1.96	–*	
DCM X allele (x) freq	0.5	0.4	0.3	0.2	
6.	0.29**	3.22**	–	0.27**	3.34**	–*	0.24**	3.55**	–**	0.19**	4.08**	–**	
Notes.

* significant at 5% level

** significant at 1% level

Table 10 Odds ratios of each PDK4 allele with χ2 significance.

Model	PDK4 allele odds ratio	
	Wt	Del	Wt	Del	Wt	Del	Wt	Del	
Individual loci	0.81	1.23							
1.	0.17**	5.84**							
2.	0.16**	6.22**							
DCM allele freq	0.5	0.4	0.3	0.2	
3.	0.19**	5.37**	0.2**	4.91**	0.22**	4.65**	0.22**	4.57**	
4.	0.52*	1.94*	0.43**	2.32**	0.34**	2.94**	0.22**	3.91**	
5.	0.74	1.36	0.71	1.36	0.66	1.51	0.59	1.69	
DCM X allele (x) freq	0.5	0.4	0.3	0.2	
6.	0.37**	2.71**	0.36**	2.76**	0.35**	2.94**	0.32**	3.1**	
Notes.

* significant at 5% level

** significant at 1% level

Selecting the most realistic model

For a model to be considered plausible, it should predict similar numbers of affected and unaffected individuals at each genotype as observed in Mausberg et al. (2011) and Owczarek-Lipska et al. (2013), predict similar DCM frequency as reported in the population (Wess et al., 2010), and give odds ratios of genotypes and alleles similar to those from the studies which report an association. To assist in determining which models meet these requirements, Table 12 shows which conditions each model meets (Tables S3–S6). From this it is possible to see that no model meets all the conditions, but two similar models, the models incorporating the two identified loci and an additional X-linked DCM locus with the novel DCM allele frequency at 0.4 and 0.5, meet all but one condition each. An additional exploration of the additional X-linked DCM allele frequency indicates that an X-linked DCM allele frequency between 0.4 and 0.5 leads to all conditions being met.

Table 11 Odds ratios of each Chr5 SNP allele with χ2 significance.

Model	Chr5 allele odds ratio	
	T	C	T	C	T	C	T	C	
Individual loci	0.15**	6.64**							
1.	0.08**	12.33**							
2.	0.13**	7.49**							
DCM allele freq	0.5	0.4	0.3	0.2	
3.	0.19**	5.34**	0.19**	5.37**	0.18**	5.55**	0.16**	6.07**	
4.	0.36**	2.76**	0.28**	3.62**	0.20**	5.08**	0.16**	7.68**	
5.	0.72	1.38	0.64	1.38	0.55	1.82	0.45**	2.23**	
DCM X allele (x) freq	0.5	0.4	0.3	0.2	
6.	0.33**	3.02**	0.3**	3.28**	0.27**	5.08**	0.23**	4.35**	
Notes.

* significant at 5% level

** significant at 1% level

Table 12 Adherence to model.

Table shows whether each model (with the new DCM allele frequency indicated) meets each condition, Y the condition is met, x the condition is not met. The number of conditions not met is also indicated.

	χ 2		OR genotype	OR allele		
Model	PDK4	Chr5 SNP	DCM freq	PDK4	Chr5 SNP	PDK4	Chr5 SNP	Number of conditions not met	
Individual	–	–	–	x	Y	x	x	3	
1.	x	x	x	Y	Y	Y	x	4	
2.	x	x	x	Y	Y	Y	x	4	
3.									
0.5	x	x	Y	Y	x	Y	Y	3	
0.4	x	x	x	Y	Y	Y	Y	3	
0.3	x	x	x	Y	Y	Y	Y	3	
0.2	x	x	x	Y	Y	Y	Y	3	
4.									
0.5	x	x	x	Y	Y	Y	Y	3	
0.4	x	x	x	Y	Y	Y	Y	3	
0.3	x	x	x	Y	Y	Y	Y	3	
0.2	x	x	x	Y	Y	Y	Y	3	
5.									
0.5	x	x	x	x	x	x	x	7	
0.4	x	x	Y	x	x	x	x	6	
0.3	Y	x	Y	x	x	x	x	5	
0.2	Y	x	x	x	x	x	Y	5	
6.									
0.5	Y	Y	Y	Y	x	Y	Y	1	
0.4	Y	Y	x	Y	Y	Y	Y	1	
0.3	x	x	x	Y	Y	Y	Y	3	
0.2	x	x	x	Y	Y	Y	Y	3	

Discussion

This study used publicly available data to test the prediction that genetic models incorporating multiple factors can better explain and predict the incidence of canine DCM than those utilising a single factor. Until now, the possibility that multiple genes combine to influence DCM phenotype has been proposed, but has not yet been tested, despite an established role for multiple loci in related diseases (Ingles et al., 2005; Xu et al., 2010; Rampersaud et al., 2011; Posafalvi et al., 2013). This is the first study to investigate the combined effect of multiple factors on the predisposition to DCM. Although our models do not explain all cases of canine DCM, by combining three factors (PDK4, Chr5 TIGRP2P73097 SNP and an X-linked locus) we show that DCM incidence can be more accurately predicted (Tables 6–12). Furthermore, as noted above the PDK4 splice site deletion is not significantly associated with DCM in the European population. But in the model incorporating only the two known loci, the PDK4 variant improves the odds ratio for the Chr5 SNP. Collectively these findings indicate that models incorporating multiple factors are more effective than those incorporating a single factor. This result is important because it has implications for future studies of the genetics and management of DCM. A better understanding of the genetic basis of DCM will permit the monitoring and earlier clinical intervention of high risk individuals thus potentially improving the outcome for affected individuals.

To assess the accuracy of each model, we performed several statistical tests. For any model to be considered an accurate representation of observed data it should predict similar numbers of affected and unaffected individuals at each genotype as have been reported in the published data. It should also predict a similar DCM frequency to that found in the population. Secondly, the odds ratios of genotypes and alleles should support an association of the specific variants with DCM. The models incorporating the two known DCM loci and an additional X-linked locus with a susceptible allele frequency of 0.46 for the novel susceptible allele met all such conditions. It is important to note that this susceptible allele frequency should have been identified by the previous GWAS studies (Mausberg et al., 2011; Meurs et al., 2012). It is therefore possible that additional cases and controls are required to complete a comprehensive GWAS analysis of DCM in Dobermans to establish the function and frequency of this predicted DCM associated locus.

Most predictive models are based on either known or simulated genotypes at multiple loci (Janssens et al., 2006; Pencina, D’Agostino & Vasan, 2008). Such models do not account for known effects of genotypes or allow the inclusion of additional as yet unknown, loci. For example, in this study all individuals possessing the Chr5 CC genotype have DCM. Our methodology is unique and useful where there are multiple known and unknown factors which do not fully account for the phenotype. In particular, our approach accommodates specific gene combinations to lead to disease, rather than incremental risk factors as is the case in other predictive models (Janssens et al., 2006; Pencina, D’Agostino & Vasan, 2008). Limitations to our methodology include the number of factors that can be modelled is limited by the available data. Despite this, our methodology could be used in other situations. While many phenotypes are the consequence of multiple loci, there can be some loci which have comparatively more important contribution to the phenotype (e.g., Strange et al., 2011; Papa et al., 2013). Identifying these loci can be the first steps in predicting phenotypes (e.g., Hayes et al., 2010; Papa et al., 2013). Following the identification of loci associated with a trait, our methodology can be used to indicate what type of additional loci may be influencing the trait of interest, which may simplify the identification of additional loci.

Conclusions

There are many unknown factors involved in the aetiology of canine and human DCM. In Dobermans, we have identified multigenic effects and a possible X-linked locus as novel variables influencing DCM risk. While the PDK4 splice site deletion and the Chr5 SNP have both been tested for association with DCM in the European population of Dobermans, the combined genotype of individuals has not yet been considered (Mausberg et al., 2011; Owczarek-Lipska et al., 2013). Our model would benefit from further genotyping of Dobermans at both the PDK4 and Chr5 variants to further validate the model. Future work is also required to identify X-linked DCM loci if the model is verified for the known loci. It is also possible that the different combinations of alleles leading to DCM in the model could affect the time taken to progress from one disease stage to the next as reported by Wess et al. (2010). If validated, our model has implications for current canine breeding practices and welfare of individuals within the breed. Individuals with allele combinations more likely to develop DCM can be monitored more intensely than those with less genetic risk, and mating pairs resulting in deleterious genotypes can be avoided. This will have improve welfare by reducing the prevalence of DCM-associated alleles within the population and potentially improving the longevity of affected dogs by enabling monitoring and earlier clinical management. By utilising similar methodology, equivalent multigenic effects and possible additional loci could be identified in human DCM, giving similar benefits to those described for Dobermans.

Supplemental Information

Supplemental Information 1 PRISMA checklist

Click here for additional data file.

Table S1 The phenotype decisions, combined genotype frequencies and predicted number of individuals for each genotype combination from the model incorporating the two known DCM loci

Click here for additional data file.

Table S2 The phenotype decisions, combined genotype frequencies and predicted number of individuals for each genotype combination incorporating the two known DCM loci + 50% (males) more susceptible to DCM

Click here for additional data file.

Table S3 Phenotype decisions, combined genotype frequencies and predicted number of individuals for genotype combinations incorporating the two known DCM loci + autosomal dominant DCM susceptibility locus

Click here for additional data file.

Table S4 Phenotype decisions, genotype frequencies and predicted number of individuals for each genotype combination incorporating the two known DCM loci + an autosomal additive DCM susceptibility locus

Click here for additional data file.

Table S5 Phenotype decisions, combined genotype frequencies and predicted number of individuals for each genotype combination incorporating two known DCM loci + an autosomal recessive DCM susceptibility locus

Click here for additional data file.

Table S6 The phenotype decisions, combined genotype frequencies and predicted number of individuals for each genotype combinations

Results from the model incorporating the two known DCM loci + an additional X-linked DCM susceptibility locus, where X is normal and x is susceptible.

Click here for additional data file.

Additional Information and Declarations

Competing Interests

Author Contributions

Richard Emes is an Academic Editor for PeerJ.

Siobhan Simpson conceived and designed the experiments, performed the experiments, analyzed the data, contributed reagents/materials/analysis tools, wrote the paper, prepared figures and/or tables, reviewed drafts of the paper.

Jennifer Edwards and Richard D. Emes performed the experiments, analyzed the data, contributed reagents/materials/analysis tools, wrote the paper, prepared figures and/or tables, reviewed drafts of the paper.

Malcolm A. Cobb analyzed the data, contributed reagents/materials/analysis tools, wrote the paper, reviewed drafts of the paper, funding.

Nigel P. Mongan and Catrin S. Rutland conceived and designed the experiments, performed the experiments, analyzed the data, contributed reagents/materials/analysis tools, wrote the paper, prepared figures and/or tables, reviewed drafts of the paper, funding.

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
