# Peer review of "A predictive model for canine dilated cardiomyopathy—a meta-analysis of Doberman Pinscher data"

_PeerJ, doi:10.7717/peerj.842_

## Round 0.1 · original submission · Major Revisions

Reviewers agree that more information and details are required mainly in the experimental design of the manuscript. Furthermore, some other important changes are also suggested.

Reviewer 1 ·

Basic reporting

The authors describe an attempt to develop a predictive model for DCM in Doberman Pinschers that includes different combinations of genetic risk factors. In my opinion there are not enough reliable data available that would allow the development of any meaningful models for DCM in dogs. The current work may have been based on correct theoretical assumptions and with the appropriate methodology, but I don't see the use of the results in either a breeding program or a future research effort to finally identify the elusive genetic risk factors.
Also, with no word the authors mention that Dobermans have different phases, constituting in VPCs only, or echo changes , or a combination of both.

Experimental design

I think that the GWAS leading to the identification of the PDK4 variant was not properly executed and consequently I doubt the predictive potential of the PDK4 variant.

The authors claim that an additional X-chromosomal locus better explains the available data. I have to admit my limited knowledge and experience in modeling. However, to me it did not become clear how the authors could discriminate between an additional X-chromosomal genetic risk locus and a sex-specific expression of disease (e.g. males have an earlier onset and more severe progression of disease than females). How would that be possible without pedigree data, which the authors did not have?
Also, the cited study by Wess et al showed that the prevalence is the same in male and female dogs, only the disease progression is different: males develop in the disease earlier echocardiographic changes and were therefore easier to identify in older studies, which did not use 24-hour-ECGs. Female dogs have longer only arrhythmias.

Validity of the findings

see above

Additional comments

see above

Reviewer 2 ·

Basic reporting

Much of the methods are not well explained; for example, the odds ratios that are central to the analyses are undefined.

Experimental design

It needs additional explanation - while I'm sure this makes sense to people who do these kinds of studies, others will not be able to follow the meager explanations.

Validity of the findings

I'm not qualified to opine on this.

Additional comments

2014:11:3093:0:1:REVIEW

Abstract

The authors indicate that dilated cardiomyopathy (DCM) is the second-most common cardiac disease in dogs. What is the basis for this statement? I suspect that mitral valve disease is most common, but is DCM more common than tricuspid valve disease or congestive heart failure of valvular origin?

When the authors write: “suggest a genetic basis for the known sex-disparity” is that not a tautologous statement, because sex itself has a genetic basis?

Materials and Methods

The authors write: “Odds ratios of each genotype for each model were obtained by testing each genotype against the other two combined. Odds ratios for each allele were also obtained. The significance of these odds ratios were assessed using χ2 tests. It is expected that the models will show similar odds ratio patterns and significance levels to those of the reported data.” This is an inadequate description of the statistical methods. What is the outcome? Presumably the predictor is the genotype? What is the explicit interpretation of the odds ratio? Were these crude or adjusted? What does “similar odds ratio patterns” mean when they are simply numbers reflecting associations? It is likewise unclear what “significance levels to those of the report data” means.

Results

Line 138 states: “each model was optimised to minimise the χ2 test statistic.” Apart from the fact that it is unclear what this even means, it is a method that should be explained in the Materials and Methods section of the manuscript, and not introduced in the Results section.

---

## Round 0.2 · Minor Revisions

Please, make the final edits following the reviewer suggestions.

Reviewer 2 ·

Basic reporting

No comments.

Experimental design

No comments.

Validity of the findings

No comments.

Additional comments

Thank you for making meaningful changes in response to my earlier comments. One final point: I recommend that you define an odds ratio as something other than relative likelihood; the latter is more strongly recognized as probability, so it would be more correct to define an odds ratio as the ratio of odds, with the latter defined as the ratio of likelihoods (or probabilities) under different conditions.

---

## Round 0.3 · accepted · Accept

We hope you continue to publish with us in the near future.